# Modeling Categorical Variables by Mutual Information Decomposition

**DOI:** 10.3390/e25050750

**Published:** 2023-05-04

**Authors:** Jiun-Wei Liou, Michelle Liou, Philip E. Cheng

**Affiliations:** 1Department of Electrical Engineering, Ming Chi University of Technology, New Taipei City 243, Taiwan; jwliou49908@mail.mcut.edu.tw; 2Institute of Statistical Science, Academia Sinica, Taipei 115, Taiwan; pcheng@stat.sinica.edu.tw

**Keywords:** graphical model, logistic model, log-linear model, mutual information

## Abstract

This paper proposed the use of mutual information (MI) decomposition as a novel approach to identifying indispensable variables and their interactions for contingency table analysis. The MI analysis identified subsets of associative variables based on multinomial distributions and validated parsimonious log-linear and logistic models. The proposed approach was assessed using two real-world datasets dealing with ischemic stroke (with 6 risk factors) and banking credit (with 21 discrete attributes in a sparse table). This paper also provided an empirical comparison of MI analysis versus two state-of-the-art methods in terms of variable and model selections. The proposed MI analysis scheme can be used in the construction of parsimonious log-linear and logistic models with a concise interpretation of discrete multivariate data.

## 1. Introduction

The selection of parsimonious log-linear and logistic models for discrete multivariate data is crucial for concise data interpretation based on statistical inference methods. Since the advent of linear regression, researchers developed model selection methods, such as ridge regression [1], the $C_{p}$ statistic [2], and the Akaike information criterion (AIC) [3]. As a tool for enhancing the prediction accuracy of a selected model, the AIC inspired the development of the Bayesian information criterion (BIC) [4], which maximizes posterior likelihood against a finite-mixture prior distribution. Researchers implemented conventional selection procedures with various loss-function penalty criteria to facilitate the selection of models and corresponding variables in generalized linear models (GLMs) [5,6,7,8]. However, the inclusion of AIC and BIC in statistics software packages [9] led to the selection of various sets of predictors when applying GLMs to analyzing the same dataset [10,11].

Researchers in the field of model and variable selection recently struggled with high dimensionality (numerous variables) and small (sparse) datasets. This led to the development of methods for the optimization of loss function criteria, such as Lasso, adaptive Lasso, elastic Net, Scad, and the Dantzig selector [12,13,14,15,16]. In a large-scale meta-analysis, Wang et al. [17] observed considerable variation in the cross-validation performance of model/variable selection methods.

When employing contingency tables as the core of discrete multivariate data analysis, researchers established that valid log-linear models can be used to generate valid logistic models; however, the converse is not true. The construction of hierarchical log-linear models is generally based on stepwise selection procedures, beginning with the inspection of two-way association effects, followed by the inspection of significant three-way and higher-order interaction effects. The effects of including additional parameters can be assessed using the likelihood ratio (LR) deviance or chi-square statistic, as a basic way of testing goodness-of-fit when discriminating between models with the nested structure of parameters [18,19,20,21,22]. In contingency table analysis, LR deviance is a log-likelihood measure of the geometric projection from data to a hypothetical model. By analogy, an algebraic definition of a log-linear or logistic model is equivalent to describing the geometry of orthogonal decomposition of the log-likelihood that intrinsically characterizes the association effects between variables.

The analysis of variance (ANOVA) for a linear model illustrates the orthogonal log-likelihood decomposition of normally distributed variables. A basic extension of this analysis is the mutual information (MI) identity, which introduces an orthogonal decomposition of the log-likelihood ratios of the joint distribution of variables [23]. For example, the model of independence between three variables can be decomposed into a sum of three orthogonal terms, including two two-way MI association terms and one three-way conditional MI (CMI) term. Each term defines a Kullback–Leibler (KL) divergence value between a pair of variables [24]. A two-way KL divergence value describes a log-likelihood projection from data to the model of independence between the variables, which characterizes the basic Pythagorean law of information identity [25]. The three-way Pythagorean law describes CMI as the sum of two orthogonal components: the interaction (which vanishes with normal distributions) and partial association. When dealing with a three-way contingency table, this law yields two-step orthogonal LR tests rather than a non-orthogonal combination of the Breslow–Day test [26] and the Mantel–Haenszel test [27]. Related examples can be found in [28]. The MI of a vector variable can generally be decomposed as a sum of orthogonal lower-dimensional MI and CMI terms, which present significant main and interaction effects between variables.

In this paper, we propose an MI decomposition approach to constructing log-linear and logistic models based on the multinomial distribution of a contingency table. In Section 2, we demonstrate that the basic forms of log-linear models can be expressed using MI and CMI terms. We also demonstrate the execution of a two-step LR test for conditional independence based on the Pythagorean law. In Section 3, we examine the main and interaction effects among factors by testing a series of MI identities in a clinical dataset, which includes six binary risk factors and the dichotomous stroke status of patients. We then use the observed significant effects to construct a concise log-linear model and present its graphical model. In Section 4, we present a parsimonious logistic model for predicting the stroke status through the MI deletion of redundant factors and effects. In Section 5, we outline the stepwise forward and backward MI selection procedure for indispensable predictors and construct parsimonious logistic model through inspecting an MI identity comprising significant main and interaction effects. In Section 6, we implement the proposed MI procedure for a German banking credit dataset (in the UCI Machine Learning Repository) which comprises 21 attributes of 1000 customers [7]. Section 7 presents an empirical comparison between the proposed MI analysis and two state-of-the-art methods.

To summarize, the proposed MI analysis can be used to construct parsimonious log-linear and logistic models for discrete multivariate data. It is also shown that discretization of continuous variables presents a legitimate and potential MI application to analyzing datasets that include both continuous and discrete variables. For the extended applications to GLMs, proper inference theory remains to be fully developed.

## 2. Log-Linear Model and Mutual Information

Among the GLMs, log-linear models are commonly applied to modeling contingency tables through the log link function, multinomial, and Poisson distributions. The conditional distribution of a multivariate Poisson distribution given the margins of the discrete variables characterizes the multinomial distribution of a contingency table [18,21,29]. We illustrate the essential connections between hierarchical log-linear models and mutual information decompositions in this section. Let $\left(X,Y,Z\right)$ denote a three-way *I* × *J* × *K* contingency table with cell counts $n_{ijk}=n_{X,Y,Z}\left(i,\;j,\;k\right)$, *i* = 1, …, *I*, *j* = 1, …, *J*, *k*
= 1, …, *K*, and total count $\sum_{i,j,k}n_{ijk}=N$. Denote the joint probability density function (pdf) and cell mean by $f_{X,Y,Z}\left(i,j,k\right)$ and 𝛍=($\mu_{ijk})$, respectively. The (multivariate) multinomial distribution of the table is
(1)$$Prob(n|f)=N!\prod_{ijk}\frac{f_{ijk}^{n_{ijk}}}{n_{ijk}!},$$
where $n=\left(n_{ijk}\right)$ and $\sum_{ijk}f_{ijk}=$ 1. Equation (1) can be derived from a random total count *N*, where **n** follows a multivariate Poisson sampling distribution
$$Prob\left(n|𝛍\right)=\prod_{ijk}\frac{e^{-\mu_{ijk}}\mu_{ijk}^{n_{ijk}}}{n_{ijk}!}.$$

Each cell mean $\mu_{ijk}$ $\left(=Nf_{ijk}\right)$ is a Poisson mean, and *e* is the Euler number [29]. Aside from a constant term, the log-likelihood of the Poisson multinomial model is
(2)$$L\left(𝛍\right)≅\sum_{ijk}n_{ijk}log\mu_{ijk}-\sum_{ijk}\mu_{ijk}.$$

In a hierarchical log-linear model, for example, the saturated model of three variables specifies the logarithmic factor in (2) as
(3)$$\log\mu_{ijk}=\lambda +\lambda_{i}^{X}+\lambda_{j}^{Y}+\lambda_{k}^{Z}+\lambda_{ij}^{XY}+\lambda_{ik}^{XZ}+\lambda_{jk}^{YZ}+\lambda_{ijk}^{XYZ}.$$

Model (3) explains the observed table through inspecting significant main effects, two- and three-way effects by checking LR deviance statistics between the nested models [18,19,21,30,31,32].

We now formulate a theoretically equivalent approach to inspecting hierarchical log-linear models such as (3) through the analysis of MI and illustrate the connection between the two procedures. In a three-dimensional table, the Shannon entropy defines a basic equation of joint and marginal probabilities as

*H*(X) + *H*(Y) + *H*(Z) = *I*(X, Y, Z) + *H*(X, Y, Z),
(4)
where
$$H\left(X,\;Y,\;Z\right)=-\sum_{ijk}f_{X,Y,Z}\left(i,j,k\right)\cdot \log\left[f_{X,Y,Z}\left(i,j,k\right)\right]$$
is the joint entropy. A marginal entropy such as *H*(X) is defined by analogy. Here,
$$I\left(X,\;Y,\;Z\right)=\sum_{ijk}f_{X,Y,Z}\left(i,j,k\right)\cdot \log\left\{\frac{f_{X,Y,Z}\left(i,j,k\right)}{f_{X}\left(i\right)f_{Y}\left(j\right)f_{Z}\left(k\right)}\right\}$$
is the MI between the three variables [33]. The MI is defined with a geometric notion; that is, it is the KL divergence value from the joint pdf onto the product of marginal pdfs, or the KL projection root in the hypothesized space of independence [24-25,34]. By factoring the joint log-likelihood, an orthogonal partition of the MI terms between the variables is expressed as the following MI identity

*I*(X, Y, Z) = *I*(X, Z) + *I*(Y, Z) + *I*(X, Y|Z).
(5)

A two-way MI term such as *I*(X, Z) is analogously defined with the marginal (X, Z) table. The CMI term *I*(X, Y|Z) defines the expected log-likelihood ratio for testing the conditional independence between X and Y across levels of Z. The right-hand side of (5) admits three information equivalent forms through exchanging the common variable Z with either X or Y. It is crucial that the MI of a three-way table (X, Y, Z) is decomposed as the sum of two two-way MI terms and a three-way CMI term. With multinomial log-likelihood, the sample version of (5) satisfies the same MI identity through the expression
(6)$$\begin{array}{l} 2N\hat{I}\left(X,\;Y,\;Z\right)\\ =2N\sum_{ijk}{\hat{f}}_{X,Y,Z}\left(i,j,k\right)\cdot \log\left\{\frac{{\hat{f}}_{X,Y,Z}\left(i,j,k\right)}{{\hat{f}}_{X}\left(i\right){\hat{f}}_{Y}\left(j\right){\hat{f}}_{Z}\left(k\right)}\right\}\\ =2N\sum_{ik}{\hat{f}}_{X,Z}\left(i,k\right)\cdot \log\left\{\frac{{\hat{f}}_{X,Z}\left(i,k\right)}{{\hat{f}}_{X}\left(i\right){\hat{f}}_{Z}\left(k\right)}\right\}\;\left(=2N\hat{I}\left(X,\;Z\right)\right)\\ +2N\sum_{jk}{\hat{f}}_{Y,Z}\left(j,k\right)\cdot \log\left\{\frac{{\hat{f}}_{Y,Z}\left(j,k\right)}{{\hat{f}}_{Y}\left(j\right){\hat{f}}_{Z}\left(k\right)}\right\}\left(=2N\hat{I}\left(Y,\;Z\right)\right)\\ +2N\underset{k}{\sum }\left[\underset{ij}{\sum }{\hat{f}}_{XY|Z}\left(i,j|k\right)\log\left(\frac{{\hat{f}}_{XY|Z}\left(i,j|k\right)}{{\hat{f}}_{X|Z}\left(i|k\right){\hat{f}}_{Y|Z}\left(j|k\right)}\right)\right]\left(=2N\hat{I}\left(X,\;Y|Z\right)\right)\\ =2N\left\{\hat{I}\left(X,\;Z\right)+\hat{I}\left(Y,\;Z\right)+\hat{I}(X,\;Y|Z)\right\}, \end{array}$$
where N is the total sample size. The notation ${\hat{f}}_{\mathrm{XZ}}\left(i,k\right)$ denotes the sample estimate of the joint pdf in the (*i, k*) cell, and ${\hat{f}}_{X}\left(i\right){\hat{f}}_{Z}\left(k\right)$ is the product pdf estimate under the assumption of independence. Other notations in (6) are defined by analogy. The constant factor 2N is added to yield quadratic approximation in distribution such that the scaled $\hat{I}\left(X,\;Z\right)$, $\hat{I}\left(Y,\;Z\right)$ and $\hat{I}(X,Y|Z)$ are asymptotically chi-square distributed with (*I*-1)(*K*-1), (*J*-1)(*K*-1) and (*I*-1)(*J*-1)*K* degrees of freedom (*dfs*), respectively. For ease of notation, the factor 2N will be implicitly used as in (6) but omitted in the rest of the text.

In application, the sample MI $\hat{I}(X,Y|Z)$ in (6), comparable to the deviance statistic $G^{2}\left(\left\{\mathrm{XZ},\;\mathrm{YZ}\right\}\right)$ [18], is used to test the hypothesis of conditional independence between X and Y given Z; that is, *I*(X, Y|Z)=0, which defines the log-linear model {XZ, YZ}. With this hypothesis, a remarkable Pythagorean law characterizes that I(X, Y|Z) depicts the hypotenuse of the right triangle with two orthogonal sides: one side is the three-way interaction *Int*(X, Y, Z), which defines the heterogeneous association between X and Y across the levels of Z; the other side is the partial association *Par*(X, Y|Z), which defines the homogeneous (uniform) association between X and Y across levels of Z. Specifically, the three-way CMI term of (5) can be expressed as the sum of two orthogonal components
(7)I(X, Y|Z)=Int(X, Y, Z) + Par(X, Y|Z)

We now illustrate the connection between (3) and (4), where (4) represents the expectation of the sample version of (3) except for a (normalizing) constant. The sum *H*(X) + *H*(Y) + *H*(Z) −
*H*(X, Y, Z) represents $\log\mu_{ijk}-\left(\lambda_{i}^{X}+\lambda_{j}^{Y}+\lambda_{k}^{Z}\right)$, and $I\left(X,\;Y,\;Z\right)$ represents the total association $\lambda_{ik}^{XZ}+\lambda_{jk}^{YZ}+\lambda_{ij}^{XY}+\lambda_{ijk}^{XYZ}$ in (3) using *I*(X, Z) + *I*(Y, Z) + *I*(X, Y|Z) of (5). The last statement and (7) explain that the sum $\lambda_{ij}^{XY}+\lambda_{ijk}^{XYZ}$ is exactly presented by *I*(X, Y|Z) =
*Int*(X, Y, Z) + *Par*(X, Y|Z), where the two-way term $\lambda_{ij}^{XY}$ in (3) actually represents the three-way term *Par*(X, Y|Z) [34]. This does not yield incorrect estimates because it is the unique three-way partial association term which must match the last two-way XY term. In other words, $\lambda_{ij}^{XY}$ should be interpreted as the three-way partial association between X and Y given Z when $\lambda_{ik}^{XZ}$ and $\lambda_{jk}^{YZ}$ are already in the model.

The sample analogs of the terms in (7), being the last summand in (6), satisfy the same Equation (7). In practice, the sample CMI $\hat{I}$(X, Y|Z) is the maximum likelihood estimate (MLE) of conditional independence. The MLE $\hat{Int}$(X, Y, Z) is computed using the iterative proportional fitting or the Newton–Raphson procedure [18]. The MLE of the partial association $\hat{Par}$(X, Y|Z) can be obtained by subtraction. Here, the Pythagorean law asserts that testing for conditional independence using the MLE $\hat{I}$(X, Y|Z) can be decomposed as a two-step LR test, where the usual test size α= 0.05 is replaced by two separate sizes $\alpha_{1}$ and $\alpha_{2}$, such that $\alpha =\alpha_{1}+\alpha_{2}-\alpha_{1}\alpha_{2}$ by the geometry of orthogonal decomposition. Because all three tests logically include testing for conditional independence, the law stipulates that the hypothesis of no interaction should be tested using $\hat{Int}$(X, Y, Z) with (*I*-1)(*J*-1)(*K*-1) *dfs* against the size $\alpha_{1}$ (more stringent than using α), and the hypothesis of uniform (partial) association should be tested using $\hat{Par}$(X, Y|Z) with (*I*-1)(*J*-1) *dfs* against the size $\alpha_{2}$ [28].

Formulas (5) to (7) can be straightforwardly extended to multi-way tables. Suppose that a four-way table (T, X, Y, Z) is available, the updated MI identity is


(8)
$$\begin{array}{l} I\left(T,\;X,\;Y,\;Z\right)\\ =\;I\left(X,\;Y,\;Z\right)\;+\;I\left(\left\{X,\;Y,\;Z\right\},\;T\right)\\ =\;I\left(X,\;Z\right)\;+\;I\left(Y,\;Z\right)\;+\;I\left(X,\;Y|Z\right)\;+\;\left[I\left(T,Z\right)\;+\;I\left(T,Y|Z\right)\;+\;I\left(T,X|\{Y,Z\}\right)\right]. \end{array}$$


Similar to the connection between (3) and (5), the difference of information decomposition between the usual log-linear model of all (six) two-way effects {XZ, YZ, TZ, XY, TY, TX}, and the MI association model in (8) is now explained. When the first three two-way terms {XZ, YZ, TZ} or the three two-way MI components in (8), are already in the model, the terms XY and TY actually represent partial association components *Par*(X, Y|Z) and *Par*(T, Y|Z) within the CMI terms *I*(X, Y|Z) and *I*(T, Y|Z) in (7), respectively. The last two-way term TX actually yields the component *Par*(T, X|{Y, Z}) within the term $I(T,\;X|\left\{Y,\;Z\right\})$ in (8). These illustrations show that the sum of the three interaction effects *Int*(X, Y, Z), *Int*(T, Y, Z) and *Int*(T, X, {Y, Z}) is the LR deviance statistic for testing the goodness of fit of the model with six two-way effects. If the interaction effects are significant, the corresponding graphical model may not be expressed as a union of non-overlapping cliques, which were termed indecomposable models by [35] and [36].

Meanwhile, the last three summands in the bracket of (8) accounts for the association between T and {X, Y, Z}. They can be used to explain the (logistic) regression of T on {X, Y, Z} when T is the response variable of interest. For instance, if the null hypothesis of vanishing CMI “$I(T,\;X|\left\{Y,\;Z\right\})=0$” is tested and accepted, X can be removed from a regression model. While information equivalent MI identities to (8) can be derived from interchanging the explanatory variables [23], the MI approach to building a log-linear or logistic model must be approved by testing for a valid MI identity.

## 3. MI Log-Linear Modelling

It is convenient to illustrate MI log-linear analysis using a contingency table of finite variables. The basic idea is to delete insignificant higher-order CMI and interaction effects as many as possible, such that the least number of significant lower-order effects are kept in the model. We emphasize that non-unique concise log-linear models can often be obtained from different sets of selected variables.

In this section, a clinical dataset is used to illustrate the MI approach for constructing a log-linear model. In the dataset, the brain computed tomography (CT) scans were available from 354 patients, who received a diagnosis of ischemic stroke in the middle cerebral arterial (MCA) territory, and 1518 control patients who did not have any ischemic stroke symptom at the time of CT scans [37]. The data were collected during 2006 through 2008 for a research project on the association between ischemic stroke and its risk factors, among which the calcification burden in the MCA territory was of main concern when specifying a logistic or log-linear model. The target variable was the status of MCA stroke patients (S: 1 = case; 0 = control), and the risk factors consisted of the calcification burden (C: 1 = yes; 0 = no) in the MCA territory, age (A: 1 ≥ 60; 0 < 60), gender (G: 1 = male; 0 = female), hypertension (H: 1= SBP > 140mm Hg or DBP > 90 mm Hg; 0 = otherwise), diabetes mellitus (D: 1 = fasting serum glucose level > 7.8 mmol/L; 0 = otherwise), and smoking (M: 1 = smoking over 1 cigarette/day; 0 = none). The risk factors were coded as 1 for case, 0 for control, except for age and gender.

Our goal is to assess parsimonious models for interpretation of the stroke dataset. We proceed by deleting the most insignificant high-way MI and CMI terms between the variables. Among the seven factors, factor C (calcification burden) yields the least significant MI measures with the other 6 factors. Putting aside the factor C, the next factor M has the least significant MI, followed by the factor G. Then, the remaining four factors {S, A, D, H} are significantly associated with each other. As an analog of (8), an information identity between the seven factors can be expressed as
(9)$$\begin{array}{l} \hat{I}\left(C,\;M,\;G,\;S,\;D,\;H,\;A\right)\\ =\hat{I}\left(\left\{A,\;D,\;G,\;H,\;M,\;S\right\},\;C\right)\;+\hat{I}\left(\left\{A,\;D,\;G,\;H,\;S\right\},\;M\right)\\ +\hat{I}\left(\left\{S,\;A,\;D,\;H\right\},\;G\right)\;+\hat{I}\left(\left\{D,\;A,\;H\right\},\;S\right)\;+\hat{I}\left(D,\;A,\;H\right). \end{array}$$

Note that all summands on the right-hand side of (9) are orthogonal to each other. The first summand is the MI between C and the other six factors, which can be decomposed into the sum of six orthogonal terms below. Each asterisk “*” indicates an insignificant chi-square test statistic at level α = 0.05, and values in the parentheses following each MI term are the LR (goodness-of-fit) statistic, degrees of freedom, and chi-square *p*-value, respectively.
(10)$$\begin{array}{l} \hat{I}\left(\left\{A,\;D,\;G,\;H,\;M,\;S\right\},\;C\right)\\ =\hat{I}\left(C,\;M|\{A,\;D,\;G,\;H,\;S\}\right)*\;(15.232,df\;=32,p\;=0.995)\\ +\hat{I}\left(C,\;G|\{S,\;H,\;D,\;A\}\right)*\;(9.768,df\;=16,p\;=0.878)\\ +\hat{I}\left(C,\;D|\{S,\;H,\;A\}\right)*\;(5.623,df\;=8,p\;=0.689)\\ +\hat{I}\left(C,\;H|\{S,\;A\}\right)*\;(5.057,df\;=4,p\;=0.281)\\ +\hat{I}\left(C,\;A|S\right)\;(31.449,df=\;2,p<0.001)\\ +\hat{I}\left(C,\;S\right)\;(96.972,df=\;1,p\;<0.001), \end{array}$$

Equation (10) can be summarized as
(11)$$\begin{array}{l} \hat{I}\left(\left\{A,\;D,\;G,\;H,\;M,\;S\right\},\;C\right)\;\\ =\hat{I}\left(C,\;\left\{M,\;G,\;D,\;H\right\}|\{S,\;A\}\right)*\;(35.68,df=\;60,p\;=0.995)\;\\ +\hat{I}\left(C,\;\{S,\;A\}\right)\;(128.421,df=\;3,p<0.001). \end{array}$$

In (10) and (11), it was found that $\hat{Int}$(C, A, S) (8.234, *df* = 1, *p* = 0.004) was a significant component of $\hat{I}$(C, A|S) such that the association between the calcification burden in the MCA territory and age differed between the case and control groups. Moreover, $\hat{I}$(C, A|S) (23.215, *df* = 1) was smaller than $\hat{I}$(C, A) (46.316, *df* = 1), so that $\hat{I}$(C, {S, A}) was less than the sum of $\hat{I}$(C, A) and $\hat{I}$(C, S).

By analogy, the factor M in (9) yields a similar decomposition as follows:(12)$$\begin{array}{l} \hat{I}\left(\left\{A,\;D,\;G,\;H,\;S\right\},\;M\right)\\ =\hat{I}\;\left(M,\;A|\{D,\;G,\;H,\;S\}\right)*\;(25.325,df\;=16,p\;=0.064)\\ +\hat{I}\;\left(M,\;D|\{G,\;H,\;S\}\right)*\;(12.589,df\;=8,p\;=0.127)\\ +\hat{I}\;\left(M,\;H|\{G,\;S\}\right)*\;(5.196,df=\;4,p=\;0.268)\\ +\hat{I}\;\left(M,\;\{G,\;S\}\right)\;(331.145,df=\;3,p<0.001) \end{array}$$

Here, the CMI estimate $\hat{I}$(M, {A, D, H}|{G, S}) is decomposed into the first three insignificant terms on the right-hand side of (12). The last summand $\hat{I}$(M, {G, S}) is equal to the sum of two significant terms $\hat{I}$(M, G) (314.21, *df* = 1) and $\hat{I}$(M, S|G) (16.935, *df* = 2). The latter is equal to the sum $\hat{Int}$(M, S, G) (7.224, *df* = 1) + $\hat{Par}$(M, S|G) (9.211, *df* = 1), which are smaller than $\hat{I}$(M, S) (12.903, *df* = 1). It indicates that, in a final valid log-linear model, the three-way CMI term $\hat{I}$(M, S|G) can possibly be replaced by the two-way MI term $\hat{I}$(M, S) through compensation of information from other terms.

Next, the factor G in (9) provides the following decomposition:(13)$$\begin{array}{l} \hat{I}\left(\left\{S,\;A,\;D,\;H\right\},\;G\right)\\ =\;\hat{I}\left(G,\;S|\{A,\;D,\;H\}\right)*\;(11.388,df=\;8,p\;=0.181)\\ +\hat{I}\left(G,\;H|\{A,\;D\}\right)*\;(8.695,df=\;4,p\;=0.069)\\ +\hat{I}\left(G,\;\{A,\;D\}\right)\;(32.605,df\;=3,p\;<0.001)\\ =\hat{I}\;\left(G,\;\left\{S,\;H\right\}|\{A,\;D\}\right)*\;(20.083,df\;=12,p=\;0.072)\\ +\hat{I}\left(G,\;D|A\right)\;(18.891,df\;=2,p<\;0.001)\\ +\hat{I}\left(G,\;A\right)\;(13.714,df=\;1,p\;<0.001). \end{array}$$

The summand $\hat{I}$(G, D|A) in (13) includes the significant interaction $\hat{Int}$(G, D, A) (13.529, *df* = 1), which is close to $\hat{I}$(G, A) and greater than $\hat{I}$(G, D) (8.742, *df* = 1).

Finally, the last two terms in (9) consist of the following significant components
(14)$$\begin{array}{l} \hat{I}\left(\left\{D,\;A,\;H\right\},\;S\right)\\ =\hat{I}\;\left(S,\;D|\{A,\;H\}\right)\;(22.368,df\;=4,p\;<0.001)\\ +\hat{I}\left(S,\;H|A\right)\;(71.886,df\;=2,p<\;0.001)\\ +\hat{I}\left(S,\;A\right)\;(88.586,df=\;1,p<\;0.001), \end{array}$$
and
(15)$$\begin{array}{l} \hat{I}\left(D,\;A,\;H\right)\;\\ =\hat{I}\left(A,\;H\right)\;(228.002,df=1,p<\;0.001)\\ +\hat{I}\left(D,\;H\right)\;(144.473,df=1,p<\;0.001)\\ +\hat{I}\left(D,\;A|H\right)\;(36.956,df=2,p<\;0.001). \end{array}$$

Equations (14) and (15) include significant interaction terms $\hat{Int}$(S, D, {A, H}) (19.690, *df* = 3), $\hat{Int}$(S, A, H) (13.543, *df* = 1, *p* < 0.001) and $\hat{Int}$(A, D, H) (16.797, *df* = 1). Here, $\hat{I}$(S, D|{A, H}) is slightly less than $\hat{I}$(S, D) (24.08), $\hat{I}$(S, H|A) is less than $\hat{I}$(S, H) (105.425), and $\hat{I}$(D, A|H) is less than $\hat{I}$(D, A) (66.98). However, significant interaction terms $\hat{Int}$(S, A, H) and $\hat{Int}$(A, D, H) may be indispensable, as illustrated below.

By collecting significant MI and interaction terms in (11) and (13)–(15), significant terms in (9) are included in the following decomposition
(16)$$\begin{array}{l} \hat{I}(C,\;M,\;G,\;S,\;D,\;H,\;A)\\ ≅\;\hat{I}\;\left(C,\;A|S\right)\;+\hat{I}\left(C,\;S\right)\;+\hat{I}\left(M,\;S|G\right)\;+\hat{I}\left(M,\;G\right)\\ +\hat{I}\left(G,\;D|A\right)\;+\hat{I}\left(G,\;A\right)\;+\hat{I}\left(S,\;A\right)\;+\hat{I}\left(S,\;H|A\right)\\ +\hat{I}(S,\;D|\left\{A,\;H\right\})\;+\hat{I}\left(A,\;D,\;H\right) \end{array}$$

Among hierarchical log-linear models, a class of graphical models composed of nodes and edges were attractive with simple network expressions [38]. Note that the graphs were constructed from models comprising mainly significant two-way effects (edges) between the variables (nodes), with possibly significant cliques (triangles). From the information identity (8), it is known that there are no more than six orthogonal two-way effects that can be derived from seven variables {C, A, S, M, G, D, H}. However, from among those two- and high-way effects in Equations (14)–(16), it is found that the set of six two-way effects {CS, MG, AH, DH, SH, DA} with the greatest MI estimates could only yield an under-fit model. Suppose that a valid log-linear model is given by inserting additional non-orthogonal two-way effects to the last model of six orthogonal effects. Then, the largest model of twelve significant two-way effects is the following null graphical model

LLM_0_ = {CA, CS, MS, MG, AH, DH, SH, SA, DA, SD, GA, GD},
(17)

which is still an under-fit model with LR deviance 171.577 (*df* = 108, *p* < 0.001). Consequently, the above analysis based on (16) indicates that a valid graphical model must include some three- or high-way interaction effects.

By (16), it was shown that adding the orthogonal three-way effects {ADG, SAH, ADH} to the subset {CA, CS, MS, MG, SD} of model (17) would yield a valid log-linear model

LLM_1_ = {CA, CS, MG, MS, ADG, ADH, SAH, SD},
(18)

with the deviance 127.96 (*df* = 105, *p* = 0.063). The three-way effect AD was repeated in (18), and the duplication can be avoided by regrouping the terms in (16) to yield another valid model

LLM_2_ = {ACS, GMS, ADG, DH, SAH, SD},
(19)

with deviance 126.53 (*df* = 103, *p* = 0.058). Now that LLM_1_ uses two fewer parameters than does LLM_2_, it can be regarded as the best parsimonious model for the MCA stroke data by the proposed MI analysis. Consequently, the so-called graphical model LLM_1_ must include three significant interaction terms {ADG, ADH, SAH} in (16). A modification of the standard diagram is recommended to plot the graphical model of LLM_1_, which shows three significant triangles (cliques) for three sets of three nodes (see Figure 1 below). This extends the construction and interpretation of graphical models beyond the existing literature, for example, [30,38].

To summarize, the above scheme of identifying the best parsimonious log-linear model LLM_1_ for the seven-way stroke data also leads to exhibiting significant edges and cliques (triangles) in a graphical model. Because the sample size was large relative to the number of variables, it is unnecessary to use a penalty criterion as commonly practiced in the literature.

## 4. MI Logistic Regression

As a member of GLMs, the logistic model extends classical linear models to the condition of using a binary (or discrete) response variable [39]. The covariates can be purely qualitative in nominal levels or mixed categorical and numerical variables. Recall Equation (8), given the target variable T and the regressors X, Y, and Z, a logistic model consisting of three significant main effects and YZ interaction effect can be expressed as
(20)$$\begin{array}{l} \mathrm{logit}\left[f_{T|X,Y,Z}\left(T=1│X=i,\;Y=j,\;Z=k\right)\right]\;\text{≡}\;\log\left[\frac{f_{T|X,Y,Z}(T=1|i,j,k)}{f_{T|X,Y,Z}(T=0|i,j,k)}\right]\\ =\beta_{0}+\beta_{j}^{Y}+\beta_{k}^{Z}+\beta_{jk}^{\mathrm{YZ}}+\beta_{i}^{X}. \end{array}$$

Model (20), being equivalent to the log-linear model {XYZ, TYZ, TX}, explains T using the composite of main and interaction effects for (Y, Z), in addition to the main effect X. The goodness-of-fit test for model (20) is examined using the deviance $\hat{Int}$(T, X, {Y, Z}) according to (8), where $\hat{Par}$(T, X|{Y, Z}) is significant for the main effect X. This ascertains correct main-effect parameter estimates bypassing the model of purely main effects (corresponding to the model with all possible two-way effects plus one three-way effect) according to the discussions of proper log-likelihood decomposition in (8). The MI approach to modeling logistic regression will be illustrated using the MCA stroke dataset.

Consider the stroke dataset where the target variable is the stroke status S (1 for cases and 0 for controls). The parsimonious log-linear model LLM_1_ of (18) presented the essential association between S and six risk factors. In particular, it was shown that the factor G (gender) is not needed for the illustration of S, because it is unrelated to S in model LLM_1_. Indeed, the first-step MI analysis confirmed this observation:(21)$$\begin{array}{l} \hat{I}\left(\left\{G,\;M,\;H,\;D,\;C,\;A\right\},\;S\right)\\ =\hat{I}\;\left(S,\;G|\{M,\;H,\;D,\;C,\;A\}\right)*\;(28.837,df=\;32,p=0.627)\\ +\hat{I}\left(S,\;M|\{H,\;D,\;C,\;A\}\right)\;(26.110,df=\;16,p=0.052)\\ +\hat{I}\left(S,\;C|\{H,\;A,\;D\}\right)(78.153,df\;=8,p<\;0.001)\\ +\hat{I}\left(S,\;\{H,\;A,\;D\}\right)(182.841,df\;=4,p<\;0.001). \end{array}$$

The first summand in (21) confirmed that factor G was dispensable as expected. Next, the second summand of (21) indicated that the factor M (smoking) was marginally significant conditional on the other four factors. A clinical concern about the calcification burden in the brain MCI territory could address whether M is related to the ischemic stroke. By this scenario, the relevant CMI terms can be expressed as
(22)$$\begin{array}{l} \hat{I}\left(S,\;M|\{H,\;D,\;C,\;A\}\right)\\ =\hat{Int}\;\left(S,\;M,\;\{H,\;D,\;C,\;A\}\right)\;(15.495,df=\;15,p\;=0.416)\\ +\hat{Par}\left(S,\;M|\{H,\;D,\;C,\;A\}\right)\;(10.615,df=\;1,p\;=0.001), \end{array}$$
which approves the significant partial association between S and M conditional on the factors {H, D, C, A}. Given the effect of M on S in Equation (22), the MI between S and the five risk factors {C, M, H, A, D} can be rearranged to yield the decomposed MI identity in Table 1 below.
(23)$$\begin{array}{l} \hat{I}\left(S,\;\{C,\;M,\;H,\;A,\;D\}\right)\\ =\hat{I}\left(S,\;D\right)\;+\hat{Par}\left(S,\;A|\;D\right)\;+\hat{Int}\left(S,\;D,\;A\right)\\ +\hat{Par}\left(S,\;H|\{D,\;A\}\right)\;+\hat{Int}\left(S,\;H,\;\{D,\;A\}\right)\\ +\hat{Par}\left(S,\;C|\{H,\;D,\;A\}\right)\;+\hat{Int}\left(S,\;C,\;\{H,\;D,\;A\}\right)*\\ +\hat{Par}\left(S,\;M|\{C,\;H,\;D,\;A\}\right)\;+\hat{Int}\left(S,\;M,\;\{C,\;H,\;D,\;A\}\right)*. \end{array}$$

By deleting two insignificant higher-order interactions, five main-effect (partial association) terms {M, C, H, A, D} and the interaction effect $\hat{Int}$(S, H, {D, A}) remain in the MI identity (23) as shown in Table 1. Although this four-way interaction term is significant, it was not selected into a valid model, because it is smaller than the sum of two three-way estimates $\hat{Int}$(S, A, H) and $\hat{Int}$(S, D, H) and can be replaced by these two three-way effects. In fact, Table 1 leads to the MI logistic model with significant main and interaction effects as
$$\begin{array}{l} \mathrm{logit}[f\left(S|C,\;M,\;H,\;A,\;D\right)]\;\\ =-3.584\;+\;1.653\;D+\;1.659A\;-\;1.003\;DA+\;1.689H\;-\;0.864AH\\ -\;0.763DH\;+\;0.495M+\;2.119C. \end{array}$$(24)

Model (24) was the most concise logistic model with the least significant effects, given the LR deviance 26.651 (*df* = 23, *p* = 0.271) in Table 1. Remarkably, the minimum AIC model using the five factors {C, M, H, A, D} can be found by including more interaction effects among {MA, MD, MH} in the standard procedure (cf. SAS CATMOD or SPSS logistic procedure). The minimum AIC model using significant parameter estimates gives
(25)$$\begin{array}{l} \mathrm{Logit}\;[f(S|C,M,H,A,D)]\\ =-3.824\;+\;1.895D\;+\;1.895A\;-\;1.130DA\;+\;1.664H\;-\;0.749DH\\ -\;0.841AH\;+\;1.180M\;-\;0.652MA\;-\;0.663MD\;+\;2.083C, \end{array}$$
with the LR deviance 18.97 (*df* = 21, *p* = 0.587). It is worth noting that the estimate 0.495 of the parameter M in (24) is closer to the raw-data logarithmic odds ratio 0.487 (between S and M) than the estimate 1.180 of M in the less concise AIC model in (25). This evidences that the effect of M on S was confounded by the extra interaction effects {MA, MD} in (25).

## 5. MI Variable and Model Selection

Variable and model selection in GLMs was a theme topic in statistical science. For data with moderate to high-dimensional variables, the MI stepwise forward variable selection scheme that also incorporates a backward deletion scheme is formulated as the main procedure in application.

Let $\left\{X,\;T\right\}$ denote a set of categorical variables in a contingency table, where T is the response variable (target) of interest and $X=\left\{X_{1},\cdots ,X_{m}\right\}$ is the set of m explanatory variables of T. A basic form of Equation (8) is the MI identity
(26)$$\begin{array}{l} I\left(X,\;T\right)\\ =I\left(X_{1},\;T\right)+I\left(X_{2},{\;T|X}_{1}\right)+\ldots +I\left(X_{m},{\;T|X}_{1},\;\ldots ,{\;X}_{m-1}\right)\\ =I\left(X_{1},\;T\right)+Int\left(X_{2},\;T,{\;X}_{1}\right)+Par\left(X_{2},{\;T|X}_{1}\right)+\ldots \\ +Int\left(X_{t},\;T,\;\left\{X_{1},\;\ldots ,{\;X}_{t-1}\right\}\right)+Par\left(X_{t},\;T|\left\{X_{1},\ldots ,{\;X}_{t-1}\right\}\right)+\ldots \\ +Int\left(X_{m},\;T,\;\left\{X_{1},\;\ldots ,{\;X}_{m-1}\right\}\right)+Par\left(X_{m},{\;T|X}_{1},\;\ldots ,{\;X}_{m-1}\right), \end{array}$$
which measures the association between the target T and **X**.

### 5.1. Forward Selection

A stepwise forward variable selection procedure is defined using the MI ratio (MIR), which is the ratio of an MI or CMI estimate to its degrees of freedom, *dfs*. When the sample size is relatively small compared to number of variables, the Bayesian method and other penalty criteria could be considered for selecting variables; for example, Lasso [15]. In this paper, we discuss the selection procedure and leave further work on sparsity to future studies.

**Step 1: Select significant predictors**. For ease of exposition, denote the final selected predictor set of *k* members as $\left\{X_{1},\cdots ,X_{k}\right\}$. To begin, denote the first selected predictor by $X_{1}$, which yields the maximum MIR estimate (with the target T) among all candidate predictors in **X**, that is,
(27)$${\mathrm{MIR}}_{1}=max_{X_{i}\in X}\left\{\frac{\hat{I}\left(X_{i},T\right)}{df\left[\hat{I}\left(X_{i},T\right)\right]}\right\}.$$

The sample MI estimate $\hat{I}\left(X_{i},T\right)$ in (27) approximates the chi-square distribution with degrees of freedom *df*$\left[\hat{I}\left(X_{i},T\right)\right]$ under the multinomial distribution, which is required to be significant. For t≥1, let $X^{\left(t\right)}=\{X_{1},\cdots ,X_{t}\}$ denote the set of selected predictors at the $t^{th}$ stage. The procedure selects a new predictor $X_{t+1}$ that offers the greatest (or most significant) MIR estimate
(28)$${\mathrm{MIR}}_{t+1}=max_{X_{i}\in \left\{X\backslash X^{\left(t\right)}\right\}}\left\{\frac{\hat{I}\left(X^{\left(t\right)}\cup X_{i},T\right)}{df\left[\left(X^{\left(t\right)}\cup X_{i},T\right)\right]}\right\}\equiv \frac{\hat{I}\left(X^{\left(t+1\right)},T\right)}{df\left[\hat{I}\left(X^{\left(t+1\right)},T\right)\right]}\;,$$
among the unselected predictors such that $X^{\left(t+1\right)}=X^{\left(t\right)}\cup X_{t+1}=\left\{X_{1},\cdots ,X_{t+1}\right\}$. Formula (28) is in principle equivalent to selecting a new predictor $X_{t+1}$ that offers the most significant CMI estimate $\hat{I}\left(X_{t+1},T|X^{\left(t\right)}\right)$ with the least *p* value. The forward selection scheme proceeds until no member in $X\backslash X^{\left(t\right)}$ can be included when t=k, and the set of selected predictors is denoted by $X^{\left(k\right)}=\left\{X_{1},\cdots ,X_{k}\right\}$.

It is possible that two candidates of a new predictor $X_{t+1}$ (for some t≥1) exist with two close (significant) MIR estimates, or two CMI estimates with closely significant *p* values. Such a case indicates that there may exist two different sets of competitive predictors. In the empirical study of Section 6 and Section 7, we will present few competitive sets of predictors which lead to equally useful models.

### 5.2. Backward Deletion

The backward deletion procedure is used to remove dispensable predictors from a set of selected variables in Step 1. The MI analysis presented in Section 3 and Section 4 employs the backward deletion scheme, when the set **X** of available (or selected) predictors are specified in the analysis.

**Step 2: Delete CMI terms.** Let $X^{\left(t\right)}=\left\{X_{1},\cdots ,X_{t}\right\}$ denote the selected set of predictors at stage t of Step 1. For t≥1, suppose that a new predictor is selected to yield $X^{\left(t+1\right)}$. Find the particular $X^{\prime }$ in $X^{\left(t\right)}$ such that $\hat{I}(\;X\;\text{′},\;T|X^{\left(t+1\right)}\backslash X^{\prime })$ yields the greatest insignificant *p*-value if it exists. Then, delete this dispensable predictor $X^{\prime }$. That is, delete $X^{\prime }$ in $X^{\left(t\right)}$ by the formula
(29)$$min_{X_{j}\in X^{\left(t\right)}}\left\{\frac{\hat{I}\left(X_{j},T|\left\{X^{\left(t+1\right)}\backslash X_{j}\right\}\right)}{df[\hat{I}\left(X_{j},T\right)|\{X^{\left(t+1\right)}\backslash X_{j}\}]}\right\}=\left\{\frac{\hat{I}\left(X^{\prime },T|\{X^{\left(t+1\right)}\backslash X^{\prime }\}\right)}{df[\hat{I}\left(X^{\prime },T\right)|\left\{X^{\left(t+1\right)}\backslash X^{\prime }\right\}]}\right\}\;\;\;\;\;\;\;.$$

Continue the deletion procedure (29) with the selected set $X^{\left(k\right)}=\left\{X_{1},\cdots ,X_{k}\right\}$ of predictors in Step 1 until it stops and yields the final set of predictors. Note that the stepwise forward selection (5.3) of Step 1 can be processed with the stepwise backward deletion (29) simultaneously to accomplish the selection scheme.

### 5.3. Model Construction

Finally, the MI model construction scheme is defined as follows.

**Step 3: Delete interaction terms.** Assume that the final set of predictors $\left\{X_{1},\;\ldots ,\;X_{m}\right\}$ are selected by Step 1 or 2. Rearrange the predictors in the MI identity (26), where k is replaced by m, such that the highest-order interaction estimate $\hat{Int}\left(X_{m},\;T,\;\left\{X_{1},\;\ldots ,{\;X}_{m-1}\right\}\right)$ is least significant (or most insignificant) based on the divided levels of significance defined by the two-step LR tests of (7). For 2≤t<k, the procedure continues with rearranging the variables $\left\{X_{1},\;\ldots ,{\;X}_{t-1},\;X_{t}\right\}$ such that $\hat{Int}\left(X_{t},\;T,\;\left\{X_{1},\;\ldots ,{\;X}_{t-1}\right\}\right)$ is least significant (or most insignificant), until evaluating the last estimate $\hat{Int}\left(X_{2},\;T,{\;X}_{1}\right)$ and stopping at the significant $\hat{I}\left(X_{1},\;T\right)$.

**Step 4: Model construction.** Use the selected predictor set $X^{\left(m\right)}$ from Step 2, to determine, by the results of Step 3, the retained significant interaction and partial association (main) effects in the logistic (or the desired regression) model. This concludes the final model selected by the rearranged MI identity examined in Step 3, which is comparable to the analysis given in Table 1 (k=6, m=5).

To summarize, the proposed MI variable selection scheme consists of Steps 1 and 2. Once the predictors are selected, Step 3 is used to identify the significant main and indispensable interaction effects such that Step 4 yields the desired (logistic) regression model. It should be emphasized again that the analysis of Table 1 has followed the rule: the interaction and partial association effects of each CMI estimate were evaluated by dividing the usual level α = 0.05 into two separate levels $\alpha_{1}$ and $\alpha_{2}$, such that the two-step LR tests based on the Pythagorean law of (7) are executed.

## 6. An Empirical Study

The Statlog (German Banking Credit Data) dataset in the UCI Machine Learning Repository (https://archieve.ics.uci.edu/ml/datasets/statlog+(german+credit+data) accessed on 27 April 2023) lists the banking data of 21 discrete attributes (categorical and discretized numerical variables) of 1000 customers (cf. [7]). This dataset was extracted from the original data of 1101 customers, in which three numerical attributes {A2, A5, A13} were discretized into discrete levels {10, 9, 5}, respectively [40]. The binary credibility variable of interest is denoted by A21≡T, (*T* = 1 for good creditworthy, *T* = 2 for not creditworthy). The other 20 attributes, denoted by the set **X**, were available to explain *T* (see Appendix A for the list of 21 attributes). Fahrmeir et al. [40] selected for the discretized data a main-effect logistic regression model with five predictors {A1, A2, A3, A4, A9} (defined as model FT_1_ in Table 2 below), which excluded “A5, credit amount” and “A13, age” from a previous selected subset of 7 attributes {A1, A2, A3, A4, A5, A9, A13}. Fahrmeir and Tutz [7] further discussed several variable selection criteria, namely the Wald test, AIC, BIC, and generalized score test, and concluded the updated analysis with the same model FT_1_ of five main effects.

Consider the MI analysis discussed in Section 5. Steps 1 to 4 were used to analyze the 1994 raw-level data with the binary target *T* and 20 discrete attributes. It yielded two MI logistic models denoted by MI_1_: {A1, A3, A12} and MI_2_: {A1, A3× A14} with the LR deviance *p*-values 0.13 and 0.08, respectively, as listed in Table 2. For this Statlog raw-level data, Table 3 records the MI decomposition of the best parsimonious model MI_1_ without significant interaction effects.

In Table 2, Fahrmeir and Tutz [7] obtained the model FT_1_: {A1, A2*, A3, A4*, A9*} for the raw-level data, which included the main-effect (subset) model FT_2_: {A1, A3, A9*}. The MI analysis showed that both models were over-fit, because each one of the three attributes {A2, A4, A9} was dispensable (marked with an asterisk “*”) given the set of predictors {A1, A3}; that is, the estimates $\hat{I}$(A9, T|A1, A3), $\hat{I}$(A2, T|A1, A3) and $\hat{I}$(A4, T|A1, A3) were insignificant. Meanwhile, the MI analysis yielded models MI_1_ and MI_2_, as shown in Table 2, using the raw-level data. The partial and interaction effects were decomposed for MI_1_ in Table 3. In Table 2, the model SIS_1_: {A1, A2*, A3} was the only model selected by the SIS method which will be explained in Section 7. In the rest of this study, the AIC estimates given in Table 2, Table 4 and Table 5 were only recorded without discussion.

## 7. Comparison with Existing Methods

Researchers recently focused on methods for the selection of models and variables to deal with data featuring high-dimensional attributes. The least absolute shrinkage and selection operator, in short, the Lasso method [15], attracted a particular attention in recent years. Examples include the smoothly clipped absolute deviation (SCAD) penalty [13], the sure independence screening (SIS) for high dimensional feature space [41], SIS for GLM [42,43], the elastic net [44], adaptive Lasso [16], the Dantzig selector [12], and feature screening with categorical data [45]. In the following, we outline a comparison of the SIS method and the proposed MI scheme in the analysis of banking credit data. The SIS computing program (http:’’CRAN.R-project.org/packageSIS accessed on 27 April 2023) yielded the unique model SIS_1_: {A1, A2*, A3} in Table 2. Note that the results exhibited over-fitting, due to the fact that the discretization of attribute A2 into 10 levels was large enough to render the CMI estimate $\hat{I}$(A2, T|A1, A3) insignificant. The same model labelled SIS_3_ in Table 5 was marginally valid (*p*-value 0.049) when A2 was discretized as a three-level attribute.

Chernoff et al. [46] proposed an innovative nonparametric approach to variable and model selection using GLMs, referred to as the “*I*-scores” method. Their method begins with the partitioning of explanatory attributes versus a target (response) variable in the data. In other words, a family of weighted mean squared differences of the target values (i.e., *I*-scores) is computed across various attribute partitions. Attributes of influence among the partitions are identified as the factors, which, if excluded (or included), would decrease (or increase) a large proportion of the *I*-scores. This means that influential attributes can be selected by evaluating sample histograms of the various *I*-scores across partitions. The authors recommended two working rules: (1) a hard thresholding (HT) rule for the selection of a predictor (effect) that induces a sufficiently large amount of changes in the *I*-score, and (2) a soft thresholding (ST) rule that induces a moderate change in the *I*-score. In practice, the attributes are partitioned into finite discrete sample spaces, such that the *I*-scores can be computed for an effective comparison. Discretizing each continuous attribute to a two- or three-level factor is a convenient feature of the *I*-score computing program at (https://github.com/adelinelo/iscreen accessed on 27 April 2023) [47].

We performed a comparison between the SIS, *I*-scores, and proposed MI methods using two modified forms of the numerical attributes A2, A5, and A13 in the original banking credit data. The first form scaled these numerical attributes as two-level attributes (e.g., {0, 1}), whereas the second form scaled them as three-level attributes (e.g., {0, 1, 2}), as shown in Appendix A. Table 4 and Table 5, respectively, present the predictors and models selected from the dataset using these two modified forms.

As shown in Table 4, the SIS method selected the main-effect model SIS_2_: {A1, A2, A3, A6*, A12*} for the first-form modified data. Model SIS_2_ was over-fit, based on the fact that the variables {A6, A12} were dispensable given {A1, A2, A3}. In other words, the CMI estimates $\hat{I}$(A6, T|A1, A2, A3) and $\hat{I}$(A12, T|A1, A2, A3) were insignificant.

Results of the *I*-Score method are listed in two parts. The first set of results includes models obtained using the hard-thresholding (HT) rule by which the *I*-Scores of each influential variable were set at ≥ 11.0 as a high threshold in the *I*-Score histogram. This yielded models HT_1_ and HT_2_. Note that as a singleton {A1}, model HT_1_ was clearly under-fit, based on the observation that all other models in Table 4 included {A1} as well as other factors to enhance data interpretation. Model HT_2_ {A1, A2, A14*, A2 × A14*} was over-fit based on the estimate $\hat{I}$(A14, T|A1, A2) = 25.39 (*df* = 16, *p* = 0.063). The second set of results provided soft-thresholding (ST) models, which allowed the selection of more attributes and parameter effects by imposing an inequality constraint (*I*-Scores ≥ 7.0) that was weaker than the HT rule. This yielded models ST_1_ and ST_2_ as presented in Table 4. Both models were over-fit, due to their use of at least four attributes and interactions (as with model SIS_2_).

MI analysis yielded two valid models for the first-form modified data, including MI_3_: {A1, A2, A12} and MI_4_: {A1, A2 × A3}. It should be noted that the modified two-level attribute A2 (*duration of individual banking credit account*) was identified as useful and was, therefore, more likely to be selected than its original 10-level version, which did not appear in Table 2.

Table 5 lists the models selected for the second-form modified data using three-level attributes of A2, A5, and A13. The unique SIS model SIS_3_: {A1, A2, A3}, which matched the over-fit model SIS_1_ (Table 2), was deemed marginally valid after modifying attribute A2 (with ten levels in Table 3, and two levels in Table 4) as a three-level attribute. This demonstrates that the poor fit of model SIS_1_ can be rectified using three-level attribute A2. The barely valid model MI_5_: {A1, A2, A12} matched the valid model MI_3_ (see Table 4); therefore, the corresponding interpretation did not vary when A2 was discretized as a two- or three-level attribute. It should be noted that model MI_6_: {A1, A3, A5} was valid when attribute A5 (*individual credit amount*) was replaced by a three-level attribute, considering that A5 never occurred in a valid model.

The *I*-Score model HT_3_: {A1, A2} proved valid (using two predictors); however, it was clearly under-fit when compared to models SIS_3_ and MI_5_. Models HT_4_, ST_3_, and ST_4_ were clearly over-fit (using four or more predictors with interaction parameters), when compared to the previous analysis of models in Table 2.

It should be noted that numerical attribute A5 (*individual credit amount*) was found to be useful (with the other two attributes A1 and A3) in explaining the target *credibility* only when applied via the discretized three-level attribute A5 in model MI_6_. This begs the question of whether model MI_6_ maintains its validity when A5 is returned to its original *numerical* status. The answer is yes—the same main-effect logistic model {A1, A3, A5 (numerical)} remains valid, with deviance of 1021.3 (*df* = 977, *p* = 0.158). The same question can be posed to the two other models: MI_3_ (MI_5_) {A1, A2, A12} and MI_4_ (SIS_3_) {A1, A2, A3} (or {A1, A2 × A3}). We found that when using the original numerical A2, model {A1, A2, A3} was indeed valid, with a deviance of 259.37 (*df* = 220, *p* = 0.354). By contrast, model {A1, A2, A12} was revised as {A1, A2, A12, A1 × A2, A1 × A12, A2 × A12}, which required the inclusion of three two-way interaction effects to yield a valid model with deviance of 234.82 (*df* = 201, *p* = 0.051).

Table 2, Table 4, and Table 5 present a summary of the selected models. Logistic regression analysis and the SIS method (Lasso type) were both shown to select a unique (i.e., best) model; however, the *I*-score method (using either hard or soft thresholding) and the proposed MI method often selected more than one valid model. It should be noted that attributes {A1, A2, A3} were consistently selected for the SIS models as well as models MI_4_, ST_1_, and ST_3_. Note, however, that among these models, only MI_4_ and SIS_3_ were deemed valid.

In the five-variable main-effect model in the original analysis by [7] and by [40], MI analysis consistently selected variables {A1, A3}, while variables {A2, A4, A9} were deemed dispensable. As shown in Table 2, the 10-level variable A2 was deemed dispensable for model SIS_1_: {A1, A2*, A3}. MI analysis confirmed that valid models could be obtained by replacing A2 with A12 (MI_1_) or A14 (MI_2_). Indeed, attributes A1 (*status of bank account*) and (two- or three-level) A2 (*length of credit account*) were selected as useful predictors by all three methods in Table 4 and Table 5.

When applied to banking credit data with a moderate sample size (1000) and moderate number of discrete explanatory attributes (20), MI analysis selected several concise logistic models by which to obtain valid interpretations of the data. Models MI_1_ and MI_2_ proved equally useful (Table 2), as did MI_3_ and MI_4_ (Table 4). MI_3_ and MI_5_ described the same model; however, they differed in their use of attribute A2 (two-level or three-level, respectively). MI analysis verified that attribute A5 in model MI_6_ was a useful predictor as a discrete three-level or numerical variable. Note that the discretization of numerical attributes in MI analysis makes it possible to obtain valid models using the same predictors in a discrete or numerical form. To summarize, MI analysis yielded parsimonious logistic models with predictors capable of explaining the *creditability* of the banking credit data.

## 8. Discussion and Conclusions

The selection of a model and corresponding variables in GLMs was an important topic in the field of statistical inference since the 1970s. Conventional approaches to inference when using contingency tables have relied on the selection of log-linear and logistic models based on AIC, BIC, and various penalty criteria. In the literature, there is no known method for the orthogonal decomposition of MI between variables that completely eliminates the collinearity issue (redundant correlation) in linear regression. MI decomposition is a valid approach to dealing with any set of continuous and discrete variables. However, it could be convenient to develop a systematic procedure by which to validate selected main-effect variables and their interaction effects (among discrete/discretized variables) by testing each selection step via standard (*p*-value) inference. In the current study, we addressed multivariate discrete data organized in the form of contingency tables. We developed a scheme that involves the construction of valid log-linear models by selecting significant MI and CMI effects of attributes, while discarding the insignificant ones.

Selecting a valid logistic model for a discrete response variable involves the stepwise forward selection of variables in conjunction with the backward deletion of redundant variables for use in validating the final set of predictors. Main and interaction effects are then identified by testing proper MI identities of the selected predictors to facilitate the construction of parsimonious models. In simulations using data related to ischemic stroke (Section 3 and Section 4), the proposed method was shown to produce log-linear and logistic models that were inherently parsimonious without the need for penalty criteria. As shown in Table 3, Table 4 and Table 5, MI analysis made it possible to construct parsimonious logistic models directly from a small set of predictors. Our analysis also revealed that state-of-the-art model/variable selection schemes failed to select a few numbers of valid competitors.

It should be emphasized that the MI model/variable selection scheme in Section 5 is recommended for the general analysis of data with an arbitrary number of attributes and a dataset of sufficient size. Note that when dealing with a sparse contingency table (a moderately large number of attributes with a relatively small sample size), this approach could generate multiple competing models. MI analysis is an efficient approach to obtaining the main and interaction effects of parsimonious logistic models by discretizing numerical attributes as lower-level discrete variables. Remarkably, the MI selected set of discrete (and discretized) predictors will remain in the model when the discretized attributes are replaced by their original numerical values via MI analysis of logistic models.

To summarize, this paper outlines the construction of parsimonious log-linear and logistic models as a proof of concept for the practical MI analysis of contingency tables. We posit that the proposed MI analysis scheme could be extended to deriving valid inferences for GLMs.

## Figures and Tables

**Figure 1 entropy-25-00750-f001:**
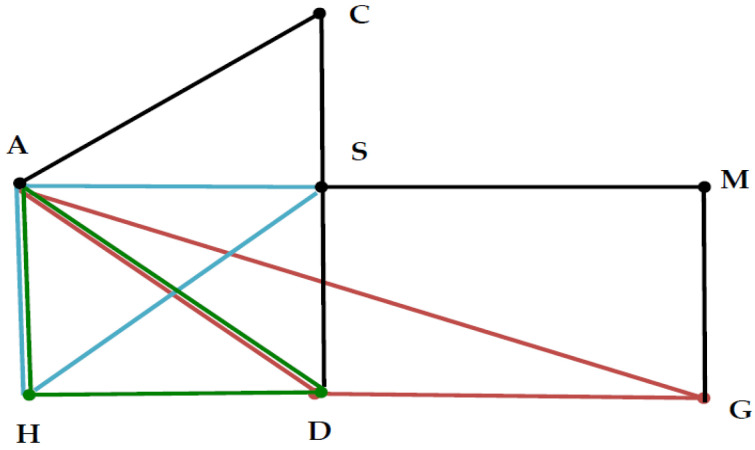
Graphical model LLM_1_: Edges {CA, CS, DS, MG, MS} and triangles {ADG, ADH, ASH} with the same corresponding colors black, red, green and cyan in the graph.

**Table 1 entropy-25-00750-t001:** Partitioned CMI terms in the MI identity (23).

Orthogonal Components	Conditional Mutual Information			Interaction			Partial Association		
	*LR*	*df*	*p*	*LR*	*df*	*p*	*LR*	*df*	*p*
$\hat{I}$(S, M|{C, H, D, A})	26.110	16	0.052	15.495	15	0.416	10.615	1	0.001
$\hat{I}$(S, C|{H, D, A))	78.153	8	<0.001	11.963	7	0.102	66.190	1	<0.001
$\hat{I}$(S, H|{D, A})	55.444	4	<0.001	12.257	3	0.007	43.187	1	<0.001
$\hat{I}$(S, A|D)	103.314	2	<0.001	27.840	1	<0.001	75.474	1	<0.001
$\hat{I}$(S, D)	24.083	1	<0.001						

**Table 2 entropy-25-00750-t002:** MI analysis of models selected by F-T, SIS, and MI for the raw-level data (an insignificant and dispensable predictor is marked with an asterisk “*”).

Method	Model	Log-Likelihood	LR (*df*) *p*-Value	AIC
FT_1_	{A1, A2*, A3, A4*, A9*}	−379.42	630.63 (5570) 1.00	816.85
FT_2_	{A1, A3, A9*}	−93.56	65.90 (112) 1.00	209.13
SIS_1_	{A1, A2*, A3}	−147.31	130.66 (183) 0.99	328.61
MI_1_	{A1, A3, A12}	−109.59	82.29 (69) 0.131	241.18
MI_2_	{A1, A3 × A14}	−79.31	55.27 (42) 0.082	194.63

**Table 3 entropy-25-00750-t003:** MI decomposition of model MI_1_ for the Statlog raw-level data.

Orthogonal Components	Conditional Mutual Information			Interaction			Partial Association		
	LR	*df*	*p*	LR	*df*	*p*	LR	*df*	*p*
$\hat{I}$(A12, T|A1, A3)	83.84	60	0.023	66.97	57	0.17	16.87	3	<0.001
$\hat{I}$(A3, T|A1)	51.28	16	<0.001	14.04	12	0.30	37.24	4	<0.001
$\hat{I}$(A1, T)	131.34	3	<0.001						

**Table 4 entropy-25-00750-t004:** Models selected by SIS, *I*-Scores and MI in using 2-level {A2, A5, 13} (Dispensable effects and under-fit models are marked with asterisks *).

Method	Model	Log-Likelihood	LR (*df*) *p*-Value	AIC
SIS_2_	{A1, A2, A3, A6*, A12*}	−250.39	339.55 (784) 1.00	532.79
*I*-Score (HT_1_, main effects only)	{A1}* *I*-Score ≥ 11.0	−10.93	131.34 (3) 0.000	29.85
*I*-Score (HT_2_) (main effects and 2-way interaction)	{A1, A2, A14*, A2 × A14*} *I*-Score ≥ 11.0	−43.15	16.33 (15) 0.361	104.30
*I*-Score (ST_1_) (main effects)	{A1, A2, A3, A6*} *I*-Score ≥ 7.0	−145.31	156.76 (187) 0.95	316.61
*I*-Score (ST_2_) (main effects and 2-way interaction)	{A1, A2 × A3, A2 × A14*, A6 × A20*} *I*-Score ≥ 7.0	−140.48	147.11 (183) 0.97	314.96
MI_3_	{A1, A2, A12}	−62.99	30.94 (24) 0.155	141.98
MI_4_	{A1, A2 × A3}	−64.96	36.52 (27) 0.104	155.91

**Table 5 entropy-25-00750-t005:** Models selected by SIS, *I*-Scores and MI methods with 3-level {A2, A5, A13} (Dispensable effects and under-fit models are marked with asterisks *).

Method	Model	Log-Likelihood	LR (*df*) *p*-Value	AIC
SIS_3_	{A1, A2, A3}	−92.98	67.65 (50) 0.049	205.96
MI_5_	{A1, A2, A12}	−92.65	54.95 (39) 0.047	203.29
MI_6_	{A1, A3, A5}	−86.08	67.16 (50) 0.053	192.17
*I*-Score (HT_3_: main effects)	{A1, A2}* *I*-Score ≥ 11.0	−27.59	4.39 (6) 0.625	67.17
*I*-Score (HT_4_: main effects and interactions)	{A1, A2, A5 × A14 × A15*} *I*-Score ≥ 11.0	−149.62	126.64 (292) 1.00	359.24
*I*-Score (ST_3_: main effects)	{A1, A2, A3, A6*} *I*-Score ≥ 7.0	−175.11	191.22 (286) 1.00	378.22
*I*-Score (ST_4_: main effects and interactions)	{A1, A2, A3 × A14*, A3 × A10*} *I*-Score ≥ 7.0	−170.46	189.92 (303) 1.00	396.92

## Data Availability

The German dataset are available online at (https://archieve.ics.uci.edu/ml/datasets/statlog+(german+credit+data accessed on 27 April 2023).

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
