# Peer review of "Modeling Categorical Variables by Mutual Information Decomposition"

_entropy, 2023, doi:10.3390/e25050750_

Round 1
Reviewer 1 Report
The manuscript is devoted to the application of the method based on Mutual Information evaluation to identify indispensable variables and their interactions in contingency table analysis. The topic of the research is interesting, the evaluation of complex objects' proximity based on mutual information can allow us to improve similar object grouping objectivity. The structure of the manuscript is correct, it contains all parts for this type of publication. However, to my mind, there are some shortcomings, which should be corrected before the manuscript accept. Below, I present my remarks.
1. The Introduction section contains both the actuality and problem statement and the literature survey. It's possible, but, to my mind, the paper will look better if this section will be divided into two parts. The Introduction will contain the actuality and importance of the problem with the allocation of the main contributions of the research at the end of this section. The Literature survey section will contain the analysis of the current research in this subject area with the allocation of unsolved parts of the general problem at the end of this section.
2. The research and results are interesting, but I think that the manuscript will look better if the authors explain in more detail the preference of the proposed method in comparison with other similar techniques analyzed in the manuscript using applied quality criteria. For example, from table 4 I see that the best model is I-score, the main effect only, etc. Please, explain the results in more detail.
In general, to my point of view, the manuscript is qualitative and it can be accepted after minor revision.
Author Response
Thanks so much for your comments on the manuscript. We made the following revision based on your comments.
1. The last two paragraphs of Introduction were rewritten. The background of the MI analysis of contingency tables was rewritten for better understanding.
2. We added the last sentence in the introduction indicating that further work remains to be carried out toward wider applications of the MI analysis. Also, this statement was also explained in the final conclusion of this manuscript.
3. The revised manuscript was also edited by a professional English editor.
Reviewer 2 Report
The authors use a Mutual information decomposition to find the most important variables when analyzing a contingency table involving ischemic stroke patients and a banking credit dataset. This approach is compared with alternative methods for constructing logistic models.
Overall, I am happy to see more work showing the utility of Mutual Information, especially in regard to finding predictive attributes. The authors correctly note variable selection when using high‐dimensional datasets is an important step in the analysis.
The paper appears to have been written for experts in statistics or even practitioners in the field (especially lines 58-74). I would suggest that this section be revised to provide more explanation for the majority of readers of the journal. The results are also described in a way that may be difficult to understand for non-experts.
I also think that some graphs or diagrams might improve the overall clarity of the presentation, particularly when comparing against the existing methods.
Author Response
Thanks so much for your comments on the manuscript. We made the following revision based on your comments.
1. The background of the MI analysis (lines 58-74) was rewritten for the ease of understanding by general readers. Also, the manuscript gave more detailed MI analysis in Section 2.
2. Thanks so much for the suggestion about a diagram for graphical log-linear models. Overlapped cliques (triangles) in a valid graphical model have not been discussed in the literature. In the revised manuscript, Figure 1 was added as a pertinent illustration.
3. The revised manuscript was edited by a professional English editor.